

# DNA-based diversity assessment reveals a new coral barnacle, *Cantellius alveoporae* sp. nov. (Balanomorpha: Pyrgomatidae) exclusively associated with the high latitude coral *Alveopora japonica* in the waters of southern Korea

Hyun Kyong Kim[1,2], Benny K.K. Chan[3], Sung Joon Song[1] and Jong Seong Khim[1]

[1] School of Earth and Environmental Sciences & Research Institute of Oceanography, Seoul National University, Seoul, Republic of Korea
[2] Animal Research Division, Honam National Institute of Biological Resources, Jeollanam-do, Republic of Korea
[3] Biodiversity Research Center, Academia Sinica, Taipei, Taiwan

## ABSTRACT

In the present study, the Indo-Pacific coral associated barnacle *Cantellius euspinulosum* (*Broch, 1931*) was found to have cryptic species in Korea, Taiwan and other regions based on molecular studies. However, the original specimens of *C. euspinulosum* from Broch have not been previously described or illustrated, making it difficult to assign which cryptic species to the original *C. euspinulosum*. The original specimen of *C. euspinulosum* was examined and illustrated here, and the species identity of *C.* cf. *euspinulosum* collected from Jejudo Island in the present study and other cryptic species (based on literature illustrations) in the Indo-Pacific were evaluated. *C. euspinulosum* from Singapore, Java, Mergui Archipelago in Andaman Sea and Nha Trang represented the *C. euspinulosum* identified by *Broch (1931)*. It is a generalist on *Acropora*, *Favia*, *Favites*, *Leptoria*, *Montipora*, *Pachyseris* and *Pocillipora* corals and distributed in the Indo-Pacific region. Morphological examination and DNA sequencing (COI, 12S DNA sequences) in the present study showed that *C.* cf. *euspinulosum* from Jejudo Island, Korea represents a distinct species, herein named *C. alveoporae* sp. nov. *Cantellius alveroporae* sp. nov. is a specialist species that only grows on *Alveopora* and also present in Palau, and Ogasawara Island in Japan. *Cantellius* cf. *euspinuloum* in Taiwan, the Moscos Island, and Australia belong to several other distinct species awaiting further morphological and molecular studies. At least five cryptic species of *C. euspinulosum* were identified in the present study, including both specialist and generalists.

# INTRODUCTION

The Indo-Pacific supports the highest marine diversity globally (*Roberts et al., 2002*). Coral reef habitats show the highest diversity in the marine environment, with most fauna

Corresponding authors
Benny K.K. Chan,
chankk@gate.sinica.edu.tw
Jong Seong Khim, jskocean@snu.ac.kr

exhibiting symbiosis with corals (*Roberts et al., 2002*; *Stella et al., 2011*; *Dreyer & Chan, 2020*). Several studies on coral symbiotic fauna have been conducted, including trapezoids and gall crabs, polychaetes, bivalves, and barnacles (*Goldberg, 2013*). Coral hosts play play significant ecological roles to coral-associated invertebrates by providing habitat, food, refuge, and mating sites (*Stella_et al., 2011*).

Barnacles (Cirripedia) are ubiquitous across marine environments (*Chan & Høeg, 2015*). Several groups of barnacles are associated with fire and scleractinian corals (*Anderson, 1992*; *Tsang et al., 2009*; *Malay & Michonneau, 2014*). These barnacle groups have different lifestyles (burrowing or epibiotic on corals), with molecular phylogeny delineating them as paraphyletic (*Malay & Michonneau, 2014*; *Simon-Blecher, Huchon & Achituv, 2007*; *Tsang et al., 2014*). The family Pyrgomatidae exclusively grows inside scleractinian corals (*Hiro, 1935*). The body cavity of pyrgomatid barnacles is embedded inside the coral skeleton, with a cylindrical base, and the shell is flattened (*Hiro, 1935*; *Chan, Wong & Cheng, 2020*). Coral-associated barnacles include generalists, which live in a wide range of corals, and specialists, which only live in specific coral host species (*Ogawa & Matsuzkai, 1992*; *Chan, Wong & Cheng, 2020*). Such host-specificity is correlated to their phylogenetic level (*Tsang et al., 2014*).

Previous biodiversity assessments of coral barnacles and their host-specificity were largely based on morphological approaches. These approaches resulted in many species (for example, *Cantellius pallidus*) being considered as generalists, showing very wide geographical distributions. Traditional morphological approaches faced challenges due to the great morphological variation exhibited by barnacles among hosts; consequently, conclusions cannot be made due to this high phenotypic plasticity (*Ross & Newman, 1973*; *Anderson, 1992*). With the development of molecular approaches in taxonomic and phylogenetic studies of crustaceans, the diversity of barnacles growing on corals has included some cryptic species. Furthermore, host ranges were found to be narrower when compared to the results deducted from morphological approaches (*Kolbasov, Chan & Petrunina, 2015*; *Kolbasov et al., 2016*). Thus, molecular approaches open up new insights on the host-usage and geographical distribution of coral barnacles.

A review by *Chan, Wong & Cheng (2020)* showed that the diversity of coral barnacles in the west Pacific region is positively associated with coral host diversity. The waters around Okinawa Island support the greatest coral diversity in the Pacific (350 species identified to date), including 20 species of coral barnacles. In contrast, only 20 coral species were recorded in the higher latitude coral community in the Japan Sea region, including four coral barnacle species (*Asami & Yamaguchi, 1997*; *Chan, Wong & Cheng, 2020*). Coral communities occurring at high latitudes are termed as marginal communities, as they are at the marginal distribution of coral triangles. Marginal coral communities are characterized by low diversity, slow growth rates, and high susceptibility to global climatic changes (*Chan et al., 2018*).

Jejudo Island in the waters of Korea is at the northern-most limit of the coral distribution in the West Pacific. In the region, only nine Scleractinian coral species have been recorded to date. These corals include the high latitude coral *Alveopora japonica*, which is a common foundation species, with its spatial distribution (and hence abundance) recently expanding

due to global climate change. *Chan et al. (2018)* studied the molecular diversity of coral barnacles in the waters around the Jejudo Island, and recorded three coral barnacle species growing on four Scleractinian coral hosts. The barnacle *Cantellius arcuatus* was only found on the coral *Montipora millipora*. However, the DNA sequences of *C. arcuatus* are similar to those of samples collected in Japan, Taiwan, Malaysia, and Papua New Guinea growing on a variety of hosts. Thus, *C. arcuatus* is likely a generalist species, with a wide distribution range. It only inhabits a single species of coral in the marginal community, probably due to the lower availability of other coral host species. Another barnacle, *Pyrgomina oulastreae* (*Utinomi, 1962*), was found on *Psammocora* and *Oulastrea* corals in the waters of Korea and Japan, suggesting it is a generalist. A third species *Cantellius* cf. *euspinulosum* was only found on *Alveopora japonica*. The DNA sequences of this barnacle species did not match any available GenBank sequences, questioning it could be a new species. However, the morphology of this species is similar to that of *Cantellius euspinulosm*.

*Cantellius euspinulosum* is an Indo-Pacific coral barnacle that invades many coral hosts (*Ross & Newman, 1973*). However, the species identity of *C. euspinulosum* remains unclear because illustrations from different reports of this species show large variations in morphological description. *Cantellius euspninulosa* was originally named *Creusia spinulosa* forma *euspinulosa* by *Broch (1931)*. Broch stated that this forma *euspinulosa* represented Darwin's *C. spinulosa* variety 1 (*Darwin, 1854*). Unfortunately, however, *Broch (1931)* did not provide any illustrations of collected forma *euspinulosa* specimens. *Ross & Newman (1973)* established the genus *Cantellius*, and considered *C. spinulosa* forma *euspinulosa* as *C. euspinulosum*. At present, all identifications of *C. euspinulosum* have been based on the drawings of opercular plates in *Darwin (1854)*. Without any clear examination of the *C. euspinulosum* from Broch's specimens, it would be difficult to ascertain whether they represent cryptic species.

The present study re-examined and illustrated the type specimen of *C. euspinulosum* from the Zoological Museum at the University of Copenhagen, and evaluated species identity and host-usage of *C.* cf. *euspinulosum* in the waters of Korea. Sequence divergence among all '*C. euspinulosum*' from GenBank sequences were also evaluated to provide a more accurate DNA-based diversity assessment on the geographical and cryptic diversity of *C. euspinulosum* in the Indo-Pacific region. As part of the study, a mini-review on global distribution of *Cantellius* species is presented and discussed with regard to management and conservation of the species diversity.

## MATERIALS & METHODS

### Field study permission
Field collections in the present study were approved under permits (No. 2436, 2016) from the National Institute of Biological Resources, Jeju Special Self-Governing Province.

### Data collection
In total, nine sites were selected in the southern waters of the Jejudo Island to sample coral-associated barnacles during August 2016 (Fig. 1). Specimens of *C.* cf. *euspinulosum* were collected through SCUBA diving at depths of 5–20 m on *Alveopora japonica* corals

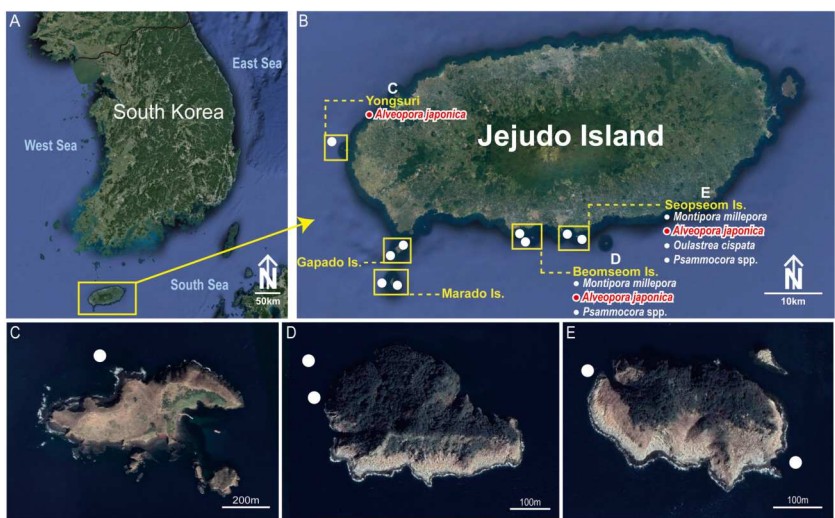

**Figure 1** **Collection sites of *Cantellius alveoporae* sp. nov. Korean Peninsula.** (A) Collection sites showing the location of the Jejudo Island; (B) Jejudo Island, showing the locations of collections sites; (C) Yongsuri; (D) Beomseom Is. and (E) Seopseom Is. Coral species were listed under each island name in (B) to show the coral distribution.

(Figs. 2A–2C). Before sampling coral-associated barnacles, the entire section of coral containing barnacles was photographed in situ to identify the host corals. Small pieces of coral with embedded barnacles (approximately 5× 5 cm) were collected using a hammer and chisel. All barnacles and host corals were preserved in 95% ethyl alcohol.

*Cantellius* cf. *euspinulosum* was isolated from *A. japonica* using forceps. The morphological characteristics of shell parts (shells, scutum and tergum) and somatic bodies (six pairs of cirri, penis, and oral cone) were examined. The shells and opercular valves (scutum and tergum) were immersed in 1.5% bleach for approximately 5 h to digest the organic tissue completely. The shells were then rinsed with slow-running purified water for 30 min and were air-dried. The shells, scutum, and tergum were gold coated and observed under Scanning Electron Microscopes (SEM; FEI Quanta 2000, Thermo-Scientific, Waltham, MA, USA). The cirri, penis, and oral cone were dissected from the somatic bodies and examined using a light microscope (Zeiss Scope A1, Zeiss, Germany) with high definition lenses (Zeiss Plan APO Chromat 40X/0.95 and ZEISS Plan APO Chromat 100x/1.4 oil). This approach allowed setal types on the cirri and mouth parts to be clearly observed. Descriptions of setals follow those of *Chan, Garm & Høeg (2008)*. Specimens were deposited in Marine Arthropods Depository Bank of Korea (MADBK) of Seoul National University (SNU) and Biodiversity Research Museum, Academia Sinica, Taiwan (ASIZCR).

## Investigation of type specimens

According to *Broch (1931)*, the type specimens of *C. epinulosum* (named as *Creusia spinulosa* forma *euspinulosa*) were preserved in five bottles. The original description on the locations of the specimens from *Broch (1931)* were detailed. First, Amboina, 3 m, coral bottom,

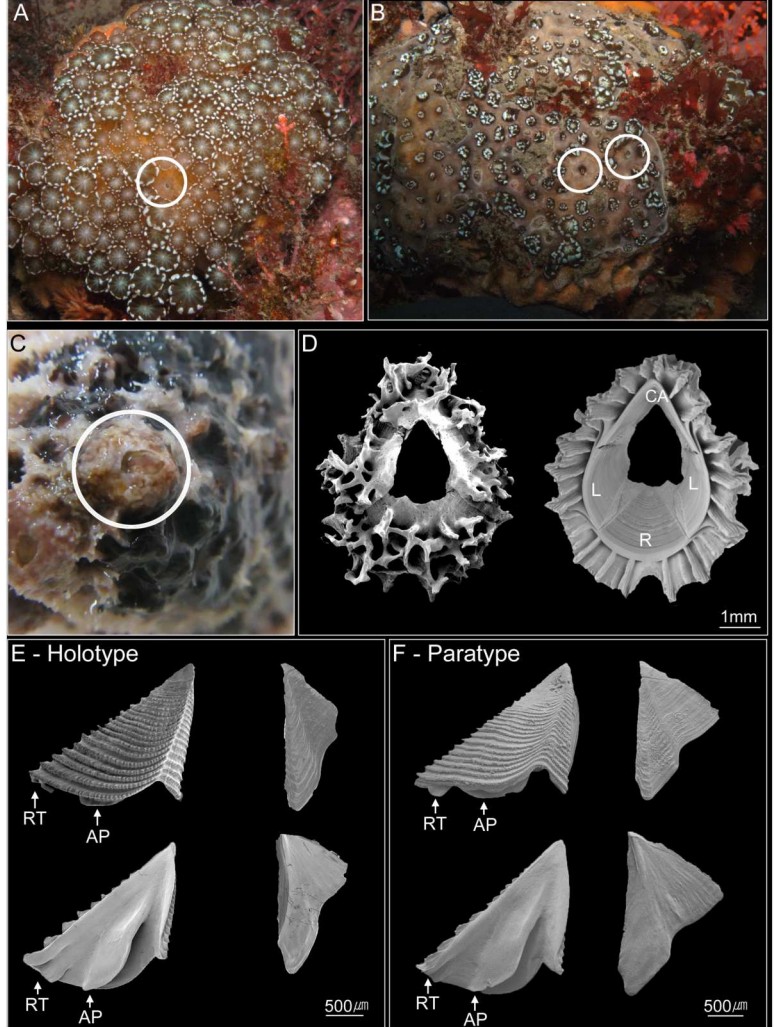

**Figure 2 Photographs of *Cantellius alveoporae* sp. nov.** (A–C) In-situ underwater photographs of *Cantellius alveoporae* sp. nov. barnacles on the host coral *Alveopora japonica* (Jejudo Island, Korea), indicated by white circles; (D) SEM investigations on the external and internal view of the *Cantellius alveoporae* shell; (E) external and internal view of the scutum and tergum of *Cantellius alveoporae* (MADBK 420501_001, Holotype); (F) external and internal view of the scutum and tergum of *Cantellius alveoporae* (ASIZCR-000455, Paratype). RT, Rostral Tooth; AP, Adductor Plate.

February 10, 1922; several specimens, mostly dead, on the lower side of the Madreporian *Herpetolitha* sp. Second, Amboina Bay, about 140 m, stones, February 22, 1922; several small specimens on dead, corroded Madreprorarian corals. Third, St. 24.5°37′S, 132°56′E, 100 m, hard bottom, April 15, 1922; several specimens in Madreporarian corals. Fourth, Singapore at low water (Consul Sc. Gad), June 14 1904; one specimen imbedded in the swollen distal end of a branched Madreporarian coral. Fifth, off Jolo, 50 m, coral and Lithothamnia (Dr. Th. Mortensen), March 19, 1914; several specimens in different Madreporarian corals (Figs. 3 and 4). All five specimens described by *Broch (1931)* were

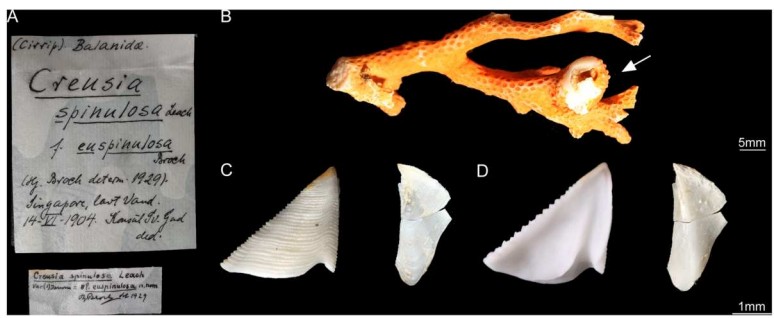

**Figure 3** *Cantellius euspinulosum* **specimen from Broch's collection, Zoological Museum, University of Copenhagen, Denmark.** (A) Specimen label of the specimens described by Broch and with Broch's signature; (B) a piece of host coral, Madreporarian coral, with one *Cantellius euspinulosum* barnacle specimen attached to it; (C) external view of the scutum and tergum of *Cantellius euspinulosum*; (D) internal view of the scutum and tergum of *Cantellius euspinulosum*.

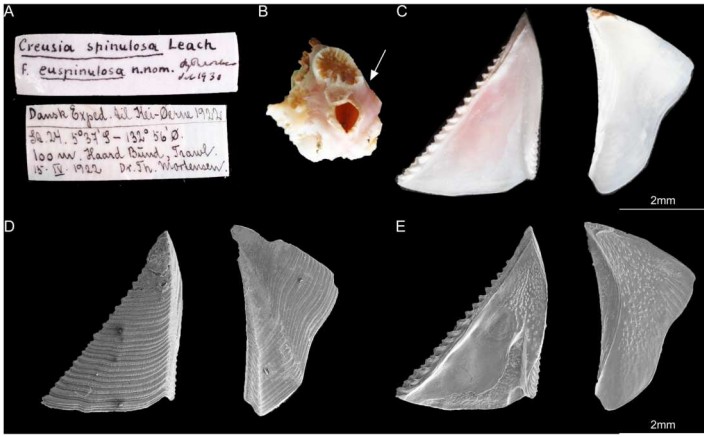

**Figure 4** *Cantellius euspinulosum* **the second specimen from Broch's collection, Zoological Museum, University of Copenhagen, Denmark.** (A) Specimen label of the specimens described by Broch and with Broch's signature; (B) a piece of host coral, Madreporarian coral with one *C antellius euspinulosum* barnacle specimen attached to it; (C) internal view of the scutum and tergum of *Cantellius euspinulosum*; (D) scanning electron micrograph showing the external view of the scutum and tergum of *Cantellius euspinulosum*; (E) scanning electron micrograph showing the internal view of the scutum and tergum of *Cantellius euspinulosum*.

successfully located in the Zoological Museum, University of Copenhagen, Denmark. All specimens were dissected to examine the scutum and tergum.

## Phylogenetic analyses

COI and 12S sequences of the *Cantellius* cf. *euspinulosum* from Jejudo Island, Korea was already uploaded in GenBank from our earlier works and used to represent the new species from the present study (*Chan et al., 2018*).

There are many mitochondrial gene sequences from coral-associated barnacles available in GenBank. These mitochondrial markers, presented as *C.* cf. *euspinulosum*, make it

possible to compare species diversity data from the waters around the Jejudo Island with other available mitochondrial sequences of *C. euspinulosum* globally. We downloaded all *Cantellius* sequences obtained from the search query "*Cantellius*" in GenBank (Table S1).

DNA sequences were proofread using MEGA version 7.0 (*Kumar, Stecher & Tamura, 2016*), and were aligned with the *Cantellius* sequences from GenBank through multiple alignment using MAFFT version 6.717 (*Katoh et al., 2002*). Alignments were also checked thoroughly, and ambiguous positions were adjusted manually. Gaps were inserted in some sequences due to missing data. A matrix of genetic distances within and among species was generated using Kimura's two-parameter model in MEGA version 7.0. The stability of clades was evaluated using bootstrap tests with 1,000 replications. A maximum likelihood (ML) test was conducted for concatenated datasets (mitochondrial COI + 12S). ML analysis was performed using RAxML-HPC2 on XSEDE (*Stamatakis, 2014*) through the online server Cyberinfrastructure for Phylogenetic Research (CIPRES) with the GTRGAMMA model of nucleotide substitution and 1,000 bootstrap replicates. In the multigene analysis, alignments of two genes were concatenated, and were then partitioned to gene regions. For the analysis, seven pyrgomatid species were used as the outgroup, and were obtained from GenBank (Table S1).

### Zoobank registration

The electronic version of this article in Portable Document Format (PDF) will represent a published work according to the International Commission on Zoological Nomenclature (ICZN), and hence the new names contained in the electronic version are effectively published under that Code from the electronic edition alone. This published work and the nomenclatural acts it contains have been registered in ZooBank, the online registration system for the ICZN. The ZooBank LSIDs (Life Science Identifiers) can be resolved and the associated information viewed through any standard web browser by appending the LSID to the prefix http://zoobank.org/. The LSID for this publication is:urn:lsid:zoobank.org:pub:4FCF5CC1-BC6B-4F22-A905-B9E1748E646C. The online version of this work is archived and available from the following digital repositories: PeerJ, PubMed Central and CLOCKSS.

## RESULTS

### Specimens of *Cantellius euspinulosum* from *Broch (1931)*

The five bottles of specimens named "*Creusia spinulosa* forma *euspinulosa*" from *Broch (1931)* contained multiple coral-associated barnacle species on examination of the shell and opercular plates. The specimens from "Amboina, 3 m", coral bottom. February 10th, 1922. Several specimens, mostly dead, on the lower side of the Madreporian *Herpetolitha* sp." (first description in section 2.2) were identified as *Pyrgomina* sp. This species has a fused external shell, with balanoid type opercular plates. The specimens from "Off Jolo, 50 m" (fifth description in section 2.2), coral and *Lithothamnia*, (Dr. Th. Mortensen). March 19th, 1914. Several specimens in different Madreporarian corals." were identified as *Armatobalanus* sp. This species has six shell plates and balanoid type opercular plates. The bottle labelled with "Amboina Bay, about 140 m" (second description in section 2.2),

stones, February 22nd, 1922. "Several small specimens on dead, corroded Madreprorarian corals." did not contain any barnacle specimens for examination.

Only two bottles of specimens were identified as *C. euspinulosum*. These included specimens from "St. 24.5°37′S, 132°56′E, 100 m, hard bottom. April 15th, 1922. Several specimens in Madreporarian corals" and "Singapore, at low water (Consul Sc. Gad). June 14th, 1904. One specimen imbedded in the swollen distal end of a branched Madreporarian coral." (Figs. 3 and 4). The scutum of both specimens was almost an equilateral triangular shape; external surfaces with horizontal striations; occludent margin straight with fine teeth, tergal margin slightly convex; basal margin convex, without any adductor plate and rostral tooth. A notch was present at 1/3 of margin close to the tergal margin, with a deep adductor muscle scar. The tergum had a slightly concaved scutal margin and wide and blunt spur. Basal margin was slightly concaved (Figs. 3 and 4).

## Molecular analysis

For the molecular analysis based on two genes (COI and 12S) sequences, 21 *Cantellius* species including seven undetermined species (*Cantellius* sp.1-7) were downloaded from GenBank. The adjusted alignments consisted of 639 bases for COI and 439 bases for 12S. To identify each species accurately, we used concatenated tree (COI + 12S) datasets. Around 15% of partial sequences of the COI and 1.8% of 12S sequences were treated with gaps due to missing data. These datasets were confirmed as 16 distinct taxa identified through clustering with previously reported species (Fig. 5). Among them, two taxa were still undetermined and considered to be new species (*Cantellius* sp.1 from Taiwan and Philippines and *Cantellius* sp.7 from Philippines). In the *C. arcuatus* clade, 76 specimens were retrieved from GenBank, which originated from Japan, Malaysia, Philippines, Papua New Guinea, and Korea. Of these, one ambiguously labelled as "*Cantellius* sp.2" from the Philippines was confirmed as *C. arcuatus*. Three GenBank submissions that were initially annotated in GenBank as *C. sextus* from Taiwan formed a unique clade. One specimen (UF8664) from the Philippines was grouped with *C. septimus* (sept1-3), as it had high sequence similarities, and we defined it as *C. septimus*. The new *Cantellius* species in the present study, *Cantellius alveoporae* sp. nov., based on nine GenBank sequence submissions originally labeled as *C. cf. euspinulosum* from Korea, were consistently clustered as distinct sister clades to the other two *Cantellius* species (*C. brevitergum* and *C. septimus*), with high nodal support that was clearly demarcated. Therefore, we named them as "*C. alveoporae* sp. nov." (see taxonomic description in section 3.3) in this study. *Cantellius* sp.5 (UF6541) and *Cantellius* sp.6 (UF8676) from the Philippines formed a sister clade with *C. transversalis* (CEL_SU46_1) and *C. acutum* (CEL_GI164_1, Taiwan and actu1), respectively. Therefore, we defined them as *C. transversalis* and *C. acutum*. *Cantellius* sp.2 (UF8663, Philippines), forming a monophyletic group with *C. arcuatum* (CEL_KT26_3 and CEL_KT101_3, Taiwan and arc1) and *Cantellius* sp.3 (UF8638, Philippines), grouped with *C. euspinulosum* (CEL_KT14_1, CEL_KT16_1 and CEL_KT27_3, Taiwan). These two groups were designated as *C. arcuatum* and *C. euspinulosum*. The remaining five groups were monophyletic: *C. pallidus*, *C. secundus*, and *C. hoegi* originating from Taiwan, *C. cf. sumbawae* and *C. iwayama* (Fig. 5).

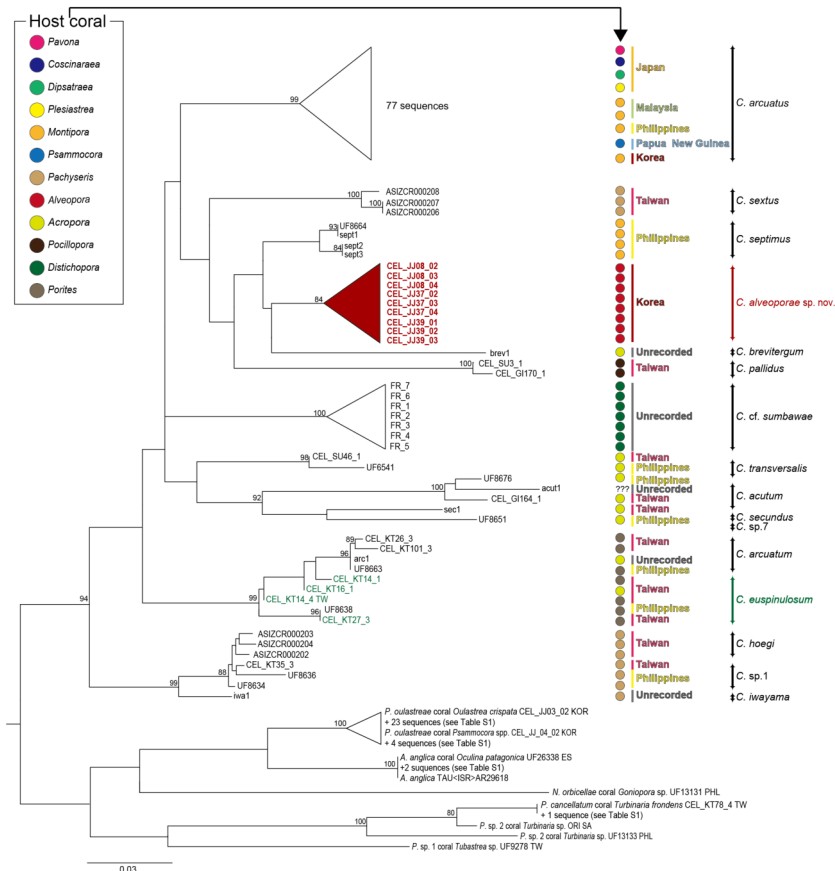

**Figure 5 Phylogenetic tree (COI + 12S) for coral-associated barnacles based on maximum likelihood (ML) analysis of the two mitochondrial gene datasets.** ML trees were constructed with RaxML 8.0.2 using the GTRGAMMA model of evolution and 1,000 bootstrap replicates. Bootstrap scores of >80% are shown at the nodes. *Cantellius euspinulosum* species reported in Taiwan are presented in green, and new species from this study are presented in pink. The scale bar represents the number of expected nucleotide substitutions per site.

Multigene sequence alignment revealed that the K2P distance among sequences ranged from 3.28% to 10.24% in *Cantellius* spp. (Table S2). Of these, the new species in the present study, *Cantellius alveoporae* sp. nov. (CEL_JJ08_02-04, CEL_JJ37_02-04, CEL_JJ39_01-03) embedded in *A. japonica*, formed a sister clade with *C. brevitergum* with high sequence divergences (Fig. 5). Their distant relationship was supported by the K2P distance, which was 5.27% (Table S2).

*Cantellius alveoporae* sp. nov. had similar morphology to *C. euspinulosum*. However, phylogenetic analysis showed that it was not likely conspecific. Four sequences from GenBank, labelled as *C. euspinulosum* originating from Taiwan formed a sister clade with *C. arcuatum*. The pairwise distance was 6.08% (Fig. 5, Table S2).

## Description

*Cantellius alveoporae* sp. nov.

*Cantellius* cf. *euspinulosum* Chan et al., 2018: 5,

Figs. 3–5.

**Zoobank registry.** urn:lsid:zoobank.org:act:F2A21CA9-CCBE-4C60-8B4A-98E3545792 E5

**Materials examined.** *Holotype.* Beom Seom, Is., Jejudo Island, Korea, on host coral *Alveopora japonica*, coll. B.K.K. Chan, 10 August 2016 (MADBK 420501_001). *Paratypes.* numerous specimens, Yongsuri, Hangyeong-myeon, Jeju-si, Jejudo Island, Korea, on host coral *Alveopora japonica*, coll. K. S. Choi, 9 August 2016 (ASIZCR-000455); 5 specimens, Beom Seom, Is., Jejudo Island, Korea, on host coral *Alveopora japonica*, coll. B.K.K. Chan, 10 August 2016 (MADBK 420501_002).

**Size (Holotype).** Basal diameter six mm.

**Diagnosis.** Base of shell with 29 longitudinal septa, shell conical with 4 plates. Scutum triangular, with rostral tooth and shallow adductor plate at basal margin, width almost equal to height, occludent margin straight with strong teeth; tergum flat, thin, spur blunt. Mandible with 5 teeth, fifth tooth fused with inferior angle; maxillule without notch; Labrum bilobed, 3 teeth on each crest. Serrulate setae present on cirrus I to III; cirrus IV to VI long, slender. Penis annulated, longer than cirrus VI.

**Description.** Shell conical and ovate, 4-plated (rostrum, carina, and paired laterals). External surface covered by coral tissue. Base of shell with approximately 29 longitudinal septa radiating from rim of sheath to external shell surface (8 in rostrum and carina, 6 and 7 in laterals), septa margin serrated. Orifice circular, about 2/5 length of rostro-carinal diameter (Fig. 2D).

Scutum and tergum separated, basically white, with purple color in apex region. Scutum triangular, width approximately equal to height, occludent margin straight, rostral tooth and adductor plate present. External surface with horizontal striations, striations with row of small pores. Internal view with a deep depressor muscle crest, without an oval-shaped adductor muscle scar. Tergum triangular. Spur blunt, width of basal margin of tergum equal to height of tergum. External surface with a shallow medial furrow, extending from basal margin towards apex, width of furrow increasing gradually from apex to base. External surface with horizontal striations (Fig. 2E).

Cirrus I with unequal rami, anterior ramus long, slender, with 12-segments, posterior ramus 9-segmented, bearing serrulate setae (Fig. 6A). Cirrus II anterior ramus with 9-segments, slightly longer than posterior ramus (7-segmented), bearing serrulate setae (Fig. 6B). Cirrus III anterior ramus longer than posterior ramus, 12- and 9-segmented, respectively, bearing serrulate setae, lacking small sharp teeth on base of each segment (Fig. 6C). Cirri IV–VI long, slender, rami similar in length, bearing serrulate setae. Cirrus IV with anterior ramus 30-segmented, posterior ramus 31-segmented, Cirrus V (anterior 30-segmented, posterior 31-segmented), Cirrus VI (anterior 32-segmented, posterior 31-segmented). Each intermediate segment of ramus of Cirrus IV–VI with 2-3 pairs of short simple setae (Figs. 6D–6F). Penis annulated, with scattered short simple-type setae. Pedicle with blunt basidorsal point (Figs. 6G–6I).

Maxilla ovate, with serrulate setae on margin (Fig. 7A). Mandibular palp elongated, bearing serrulate setae distally and on interior margin (Fig. 7B). Mandible with 5 teeth, excluding inferior angle. First 3 teeth occupy 4/5 length of cutting edge (Fig. 7C). Lateral

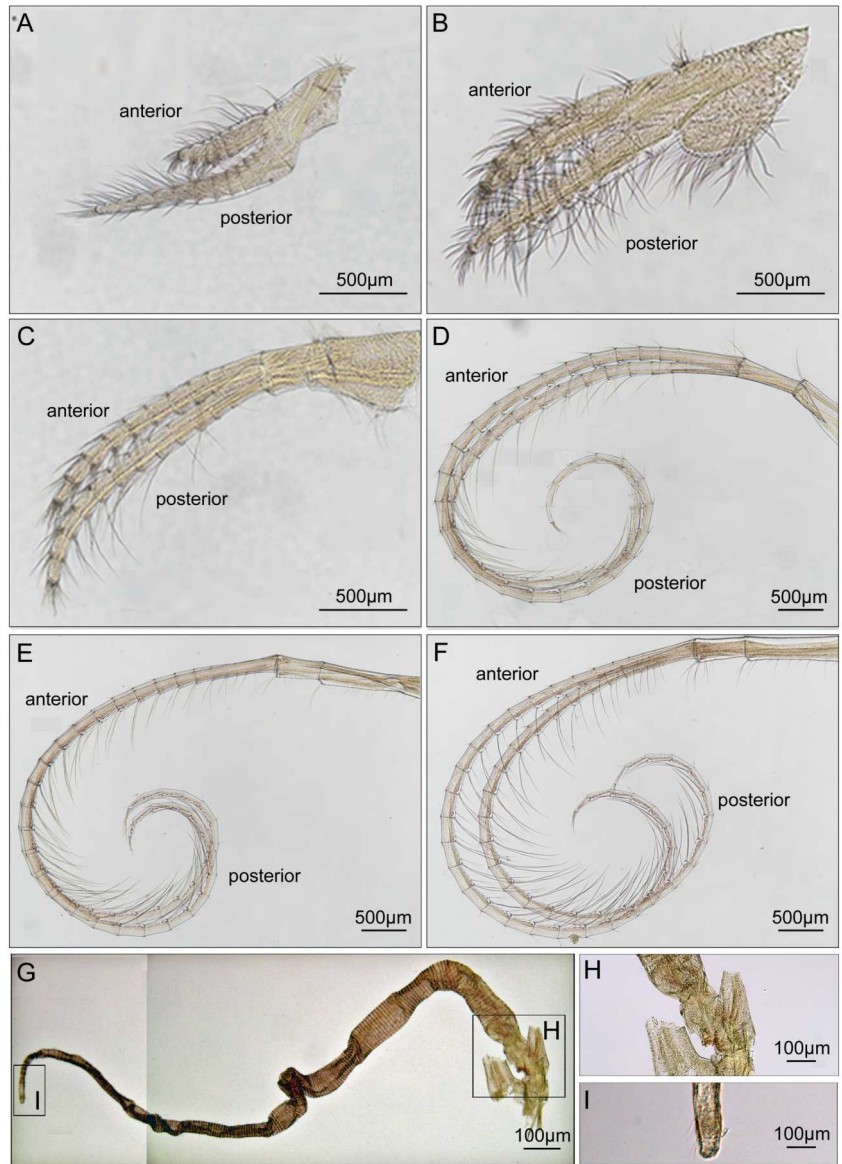

**Figure 6** **Cirri and penis of *Cantellius alveoporae* sp. nov. (MADBK 420501_001), holotype.** (A) Cirrus I; (B) cirrus II; (C) cirrus III; (D) cirrus IV; (E) cirrus V; (F) cirrus VI; (G) penis; (H) basidorsal point of penis; (I) tip of penis.

surface, lower margin and cutting edge of mandible bearing simple-type setae (Fig. 7D). Lower margin short, inferior angle blunt with simple-type setae (Fig. 7E). Maxillule cutting edge straight without notch, bearing row of 9 large setae (Fig. 7F). Region close to cutting edge with dense simple-type setae, anterior and posterior margins with simple-type setae (Figs. 7G and 7H). Labrum bilobed, lobes separated by a V-shaped notch, 3 sharp teeth on each side of notch (Figs. 7I and 7J).

**Habitat.** Grows on host coral *A. japonica*. The species located below the extended polyps of *Alveropora*. The barnacles can be located after disturbing the coral polyps to allow them
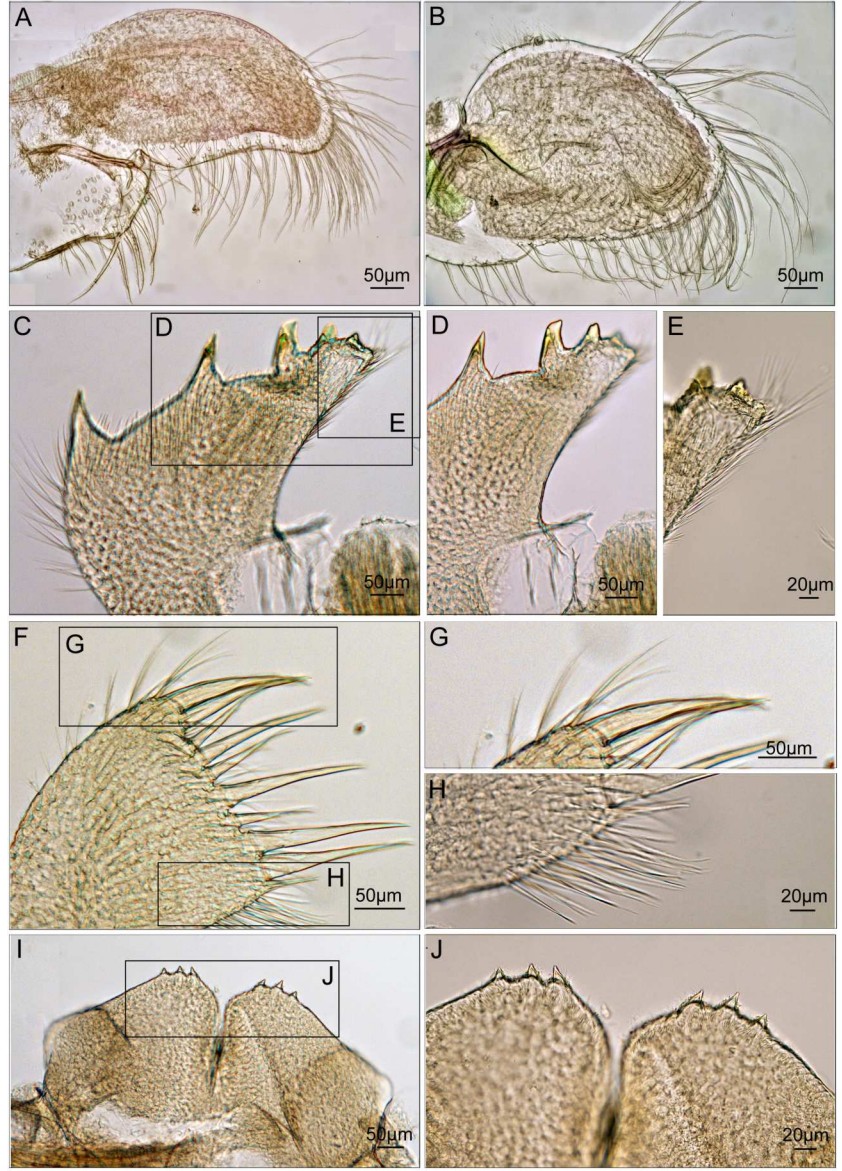

**Figure 7** **Trophi of *Cantellius alveoporae* sp. nov. (MADBK 420501_001), holotype.** (A) Maxilla; (B) mandibular palp; (C) mandible; (D) inferior margin of mandible; (E) lower margin with simple-type setae of mandible; (F) maxillule; (G) large simple- type setae on cutting edge; (H) simple-type setae on cutting edge; (I) labrum; (J) teeth on the labrum.

retract into the skeleton, exposing the barnacles when the shelter of the active polyps is removed (Figs. 2A and 2C).

**Distribution.** At present only recorded in Jejudo Island, Korea.

**Etymology.** Denotes growing on its host *A. japonica* in Korean waters.

**Remarks.** After examining the type specimens of *C. euspinulosum*, *C. alveoporae* sp. nov. differs to *C. euspinulosum* (*Broch, 1931*) in the morphological characteristics of the scutum and tergum: (1) the scutum has a rostral tooth in *C. alveoporae* sp. nov., (2) basal margin
has a shallow adductor plate in *C. alveoporae* sp. nov., (3) tergal spur sharp and thin in *C. alveoporae* sp. nov., and (4) Basal margin of tergum forms a strong V-shaped in *C. alveoporae* sp. nov. Based on these differences, the specimens from Korea were assigned as a new species.

## DISCUSSION

### Historical remarks on the *Cantellius* taxonomy

The present study reported the first examination of the type specimens of *C. euspinulosum*, providing baseline information for identifying it, along with morphologically close species. The morphology of type specimen of *C. euspinulosum* described by *Broch (1931)* matched the illustration of *Creusia spinulosa* var. 1 by Darwin. This result supports the conclusion of *Broch (1931)* that the forma *euspinulosum* represents the *Creusia spinulosa* var. 1 of Darwin. By combining morphological and molecular approaches to examine morphological variation in the opercular plates of *C. euspinulosum* and sequences divergences, we potentially discovered five cryptic species of *C. euspinulosum*. Each cryptic species had different morphological characters in the opercular plates and with respect to geographical distribution.

Based on the morphology of specimens in the opercular plates, the first group of species was considered as *C. euspinulosum* (*Broch, 1931*). This group encompassed specimens from Java, Indonesia (*Darwin, 1854*), Singapore (*Broch, 1931*), Mergui Archipelago in the Andaman Sea (*Nilsson-Cantell, 1938*), and Nha Trang, Vietman (*Poltarukha & Dautova, 2018*). This group represents the *C. spinulosa* var. 1 described by Darwin (Fig. 8). The opercular plates were similar to the type specimens examined in the present study. The scutum lacked a rostral tooth and adductor plate on the basal margin. The tergum had a broad spur. Host corals recorded by these studies included *Acropora*, *Favia*, *Favites*, *Leptoria*, *Montipora*, *Pchyseris*, and *Pocillipora* (*Nilsson-Cantell, 1938*; *Poltarukha & Dautova, 2018*). Thus, *C. euspinulosum* is likely a generalist species distributed in the Indo-Pacific region (Fig. 8A).

*Nilsson-Cantell (1938)* described the specimens of *C. euspinulosum* from the Moscos Islands in the Andaman Sea. The scutum of this species has a rostral tooth on the basal margin of the scutum, similar to the specimens from the Jejudo Island in Korea, Ogasawara in Japan, and Palau. However, this specimen had a curved spur on the tergum, contrasting with the specimens collected from Korea, Japan, and Palau. Thus, it might be another cryptic species specific to the Indian Ocean (Fig. 8B).

*Anderson (1992)* reported *C. euspinulosum* from New South Wales, Australia. The scutum of this species has a rostral tooth, but the tergum is arrow-shaped (Fig. 8C). Thus, it might represent another cryptic species present in the Pacific waters of Australia. However, to date, there are no available GenBank sequences of *Cantellius* collected from this region.

*Chan, Chen & Achituv (2013)* described *C. euspinulosum* originating from Taiwan. The illustration of the scutum showed that the basal margin does not contain an inconspicuous or minute rostral tooth; however, the depressor muscle region is wide and well-developed.

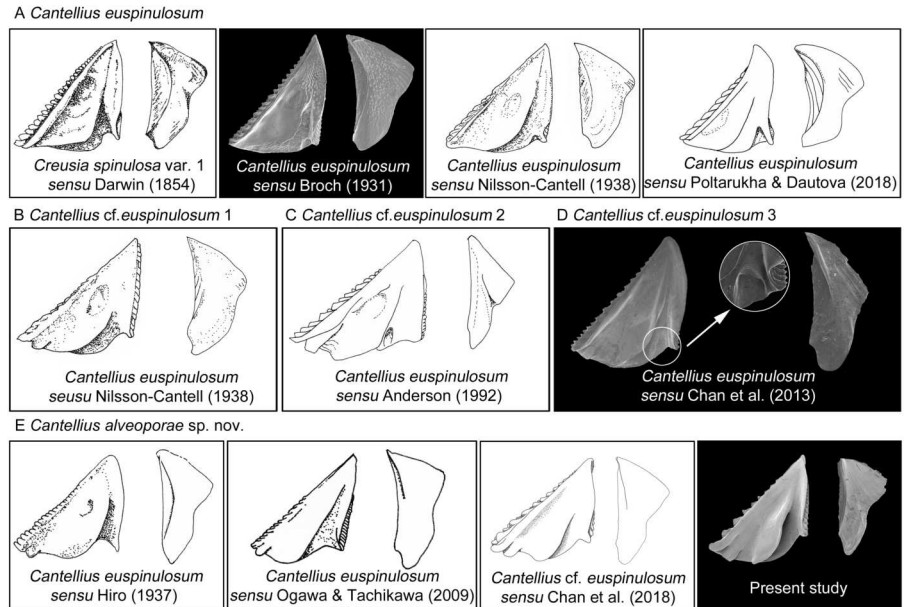

**Figure 8  Scanning electron micrographs and revised drawings from previous published Illustrations.**
(A) *Cantellius euspinulosum*; (B–D) *Cantellius* cf. *euspinulosum* 1–3; (E) *Cantellius alveoporae* sp. nov.

The species is mostly found in the coral *Porites* (*Chan, Chen & Achituv, 2013*). The presence of a separate molecular clade for this species suggests it is another cryptic species (Figs. 5 and 8D).

The last group included *C. alveoporae* sp. nov. identified around the Jejudo Island. (Fig. 8E; present study; *Chan et al., 2018*). This group might also be represented by *C. euspinuloum* reported from Palau (*Hiro, 1937b*) and Ogasawara, Japan (*Ogawa & Tachikawa, 2009*; Fig. 8E). These specimens had a rostral tooth and adductor plate on the basal margin of the scutum, and grow exclusively on the coral *Alveopora* spp. (Fig. 8E). Thus, this species is likely a specialist on *Alveopora* that is distributed in Pacific waters. Molecular studies by *Kang et al. (2020)* showed that clear genetic population differentiation exists for *A. japonica* coral among Taiwan, Korea, and Japan. Thus, these corals are likely present in all three regions at geological time scales, with limited gene flow across regions. Whether their epibiont *C. alveoporae* sp. nov. exhibits similar levels of population differentiation requires further investigation by collecting samples from different geographical regions.

## Distributions of *Cantellius* species and ecological implications

The genus *Cantellius* is only present in the Indo-Pacific region. From reviewing the global distribution records of *Cantellius* (Fig. 9), the representative species diversity could be seen from several countries encompassing Japan (13 species), Taiwan (11 species), Palau (10 species), the Philippines (7 species), and Singapore (5 species), in order. Such result might be resulted from more extensive sampling effort in some specific regions followed by fine taxonomy works on coral barnacles (*Hiro, 1935*; *Hiro, 1937a*; *Hiro, 1937b*; *Chan, Chen &*

## Global distribution of *Cantellius* species

**Andaman Island**

| Species | Ref. |
|---|---|
| C. euspinulosum | 2 |
| C. secundus | 2, 8 |
| C. transversalis | 2 |

**Hong Kong**

| Species | Ref. |
|---|---|
| C. arcuatus | 7 |
| C. secundus | 8 |
| C. pallidus | 7 |

**Taiwan**

| Species | Ref. |
|---|---|
| C. acutum | 7, 9 |
| C. arcutum | 7 |
| C. arcuatus | 7 |
| C. euspinulosum | 3 |
| C. hoegi | 7 |
| C. iwayama | 7 |
| C. pallidus | 5, 7 |
| C. septimus | 7, 8 |
| C. secundus | 7 |
| C. sextus | 5, 7 |
| C. transversalis | 7 |

**Korea**

| Species | Ref. |
|---|---|
| C. arcuatus | 9 |
| C. euspinulosum | 6, 9 |
| C. alveoporae sp. nov. | 11 |

**Japan**

| Species | Ref. |
|---|---|
| C. acutus | 3 |
| C. arcuatus | 3 |
| C. brevitergum | 3 |
| C. euspinulosum | 3 |
| C. iwayama | 3, 8 |
| C. pallidus | 2, 3 |
| C. secundus | 2, 3 |
| C. sinensis | 3 |
| C. septimus | 3 |
| C. sumbawae | 3 |
| C. septimus | 3, 8 |
| C. transversalis | 3 |
| C. tredecimus | 3 |

**Philippines**

| Species | Ref. |
|---|---|
| C. acutum | 7 |
| C. arcuatum | 7 |
| C. pallidus | 2 |
| C. secundus | 7 |
| C. septimus | 7 |
| C. sextus | 2 |
| C. transversalis | 7 |

**Palau Island**

| Species | Ref. |
|---|---|
| C. acutum | 1, 7 |
| C. arcuatum | 1, 2 |
| C. arcuatus | 1, 7 |
| C. brevitergum | 1, 2 |
| C. euspinulosum | 1, 7 |
| C. iwayama | 1, 2, 8 |
| C. secundus | 1, 2, 8 |
| C. septimus | 1, 2, 8 |
| C. sextus | 1, 2 |
| C. pallidus | 1, 7 |

**Celebes Sea**

| Species | Ref. |
|---|---|
| C. pallidus | 10 |

**Mayotte**

| Species | Ref. |
|---|---|
| C. cornutergum | 10 |
| C. pallidus | 10 |

**Java Sea**

| Species | Ref. |
|---|---|
| C. pallidus | 10 |
| C. sumbawae | 10 |

**Seychelles**

| Species | Ref. |
|---|---|
| C. alphonsei | 4 |

**Malaysia**

| Species | Ref. |
|---|---|
| C. arcuatus | 9 |

**Singapore**

| Species | Ref. |
|---|---|
| C. euspinulosum | 2 |
| C. gregarious | 7 |
| C. pallidus | 2 |
| C. secundus | 8 |
| C. tredecimus | 2 |

**Indian Ocean**

| Species | Ref. |
|---|---|
| C. secundus | 8 |
| C. septimus | 2 |
| C. transversalis | 7 |

**Banda Sea**

| Species | Ref. |
|---|---|
| C. gregarious | 2 |
| C. pallidus | 7 |

**Papua New Guinea**

| Species | Ref. |
|---|---|
| C. arcuatus | 9 |

**Australia**

| Species | Ref. |
|---|---|
| C. pallidus | 7 |
| C. septimus | 8 |
| C. secundus | 8 |

**References**

| # | Reference |
|---|---|
| 1 | Hiro (1937b) |
| 2 | Newman & Ross (1976) |
| 3 | Asami & Yamaguchi (1997) |
| 4 | Achituv (2001) |
| 5 | Achituv et al. (2009) |
| 6 | Kim (2011) |
| 7 | Chan et al. (2013) |
| 8 | Poltarukha & Dautova (2018) |
| 9 | Chan et al. (2018) |
| 10 | Zweifler et al. 2020 |
| 11 | Present study |

**Figure 9 Global distribution of *Cantellius* species, with a total of 19 species reported including three species in Korea.** For clarity, literatures in figure were represented by numericals. 1 = *Hiro, 1937b*, 2 = *Newman and Ross, 1976*, 3 = *Asami and Yamaguchi, 1997*, 4 = *Achituv, 2001*, 5 = *Achituv, Tsang & Chan, 2009*, 6 = *Kim, 2011*, 7 = *Chan, Chen & Achituv, 2013*, 8 = *Poltarukha and Dautova, 2018*, 9 = *Chan et al., 2018*, 10 = *Zweifler et al., 2020*, 11 = Present study.

*Achituv, 2013*; *Utinomi, 1962*). Meantime, climate condition and management practices on sustaining coral habitats would also be acknowledged.

All other countries showed relatively lower species diversity in *Cantellius*, with maximum of three *Cantellius* species being documented. The occurrence of at least one species was reported from the Andaman Island, Australia, Hong Kong, Indian Ocean, Korea, the Banda Sea, Malaysia, Papua New Guinea, and Seychelles. A lesser species diversity in these areas would be collectively explained by inadequate study, sampling endeavor, decreasing population of host corals, etc.

*Cantellius alveoporae* sp. nov. was identified in the present study, and is a specialist on *A. japonica* coral. *Alveopora japonica* is high latitude coral that is distributed in the waters of southern Taiwan, the Jejudo Island, Korea, and around the Honshu Island, Japan (*Veron, 2002*). This coral is an important foundation species on high latitude coral reefs, and forms secondary habitats for other species, including coral barnacles (*Noseworthy et al., 2016*). Under the effect of global climatic change, the water temperature of the west Pacific is increasing, particularly in the waters around Korea. Over the last 51 years, the water temperature in the Pacific has increased by 1.23 °C, while that around Korea is 2.5 times greater than the global trend (*Belkin, 2009*; *Han & Lee, 2020*).

Recent studies documented that current global warming might increase the recruitment and abundance of *A. japonica* around Korea (*Denis et al., 2015*; *Vieira et al., 2016*). Thus, the abundance of *C. alveoporae* sp. nov. would be also expected to increase with that of *A.*

*japonica*. Phylogenetic analysis of the coral *A. japonica* collected from the southern waters of Japan, Jejudo Island in Korea, and Taiwan showed that *Alveopora* forms three molecular clades matching these three regions (*Kang et al., 2020*). Thus, *A. japonica* in Korea would have established over a long geological history. Therefore, the coral barnacle *C. alveoporae* sp. nov. might exhibit genetic divergence in Japan, Korea, and other Pacific locations, with DNA-based assessments being required to confirm this hypothesis.

Meantime, the current global climatic changes and the bleaching events of corals in the Pacific, including Korean waters, might impact both corals and coral symbiotic fauna. As the coral reef assemblages on the Jejudo Island, Korea is marginal community, *Cantellius* diversity would be limited to the southern part of the Korean coastal waters. Accordingly, a drastic effect on the diversity of *Cantellius* would be prospected. Such impact would increase for those especially having narrower latitudinal distribution for *C. hoegi* and *C. brevitergum* which presently recorded in Taiwan and Palau only, respectively. Further studies on diversity and biogeography of coral barnacles should be conducted in the coral triangle region, and only knowing more on the detailed diversity pattern can allow better management and prediction of future diversity of coral associated barnacles in the Indo-Pacific region (*Tittensor et al., 2010*).

## ACKNOWLEDGEMENTS

We would like to thank Prof. Kwang-Sik Choi (School of Marine Biomedical Science, Jeju National University) for assisting with field collection. The authors would like to thank the reviewers, Christine Ewers-Saucedo and the other anonymous reviewer for giving constructive comments on the MS. Thanks to Danny Eibye-Jacobsen and Niklas Dreyer (Zoological Museum, University of Copenhagen) for arranging museum loans of specimens from Zoological Museum, University of Copenhagen, Denmark.

### Funding

This work was supported by the projects entitled "Ecosystem-Based Analysis and Decision-Making Support System Development for Marine Spatial Planning [grant number 20170325]" funded by the Ministry of Oceans and Fisheries, Korea. The funders had no role in study design, data collection and analysis, decision to publish, or preparation of the manuscript.

### Grant Disclosures

The following grant information was disclosed by the authors:
Ministry of Oceans and Fisheries, Korea: 20170325.

### Competing Interests

The authors declare there are no competing interests.

## Author Contributions

- Hyun Kyong Kim and Benny K.K. Chan conceived and designed the experiments, performed the experiments, analyzed the data, prepared figures and/or tables, authored or reviewed drafts of the paper, and approved the final draft.
- Sung Joon Song and Jong Seong Khim analyzed the data, authored or reviewed drafts of the paper, and approved the final draft.

## Field Study Permissions

The following information was supplied relating to field study approvals (i.e., approving body and any reference numbers):

Field collections in the present study were approved under permits (No. 2436, 2016) from the National Institute of Biological Resources, Jeju Special Self-Governing Province.

## DNA Deposition

The following information was supplied regarding the deposition of DNA sequences:

The sequencing data of the new species is available in an earlier publication reporting *Cantellius cf. euspinulosum* in PLOS ONE (*Chan et al., 2018*).

The sequences described here are available at GenBank: MG878707 to MG878715 for COI and MG878770 to MG878772, MG878825 to MG878827 and MG878831 to MG878833 for 12S.

These GenBank sequences are listed in Table S1.

## Data Availability

The sequencing data of the new species is available in our earlier publication reporting *Cantellius cf. euspinulosum* in PLOS ONE (*Chan et al., 2018*). The location of each specimen, the specimen identification/ accession number and host information are available in Table S1.

## New Species Registration

The following information was supplied regarding the registration of a newly described species:

Publication LSID: urn:lsid:zoobank.org:pub:4FCF5CC1-BC6B-4F22-A905-B9E1748E646C. *Cantellius alveoporae* sp. nov. LSID: urn:lsid:zoobank.org:act:F2A21CA9-CCBE-4C60-8B4A-98E3545792E5.

## Supplemental Information

Supplemental information for this article can be found online at http://dx.doi.org/10.7717/peerj.11284#supplemental-information.

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
