# Peer review of "DNA-based diversity assessment reveals a new coral barnacle, Cantellius alveoporae sp. nov. (Balanomorpha: Pyrgomatidae) exclusively associated with the high latitude coral Alveopora japonica in the waters of southern Korea"

_PeerJ, doi:10.7717/peerj.11284_

## Round 0.1 · original submission · Minor Revisions

I have now heard back from two expert referees who are both very supportive of this work and encourage its eventual publication. Each has a number of suggestions for how the manuscript could be improved, but these are largely editorial in nature, and I do not expect that you will have any troubles in addressing them. I look forward to seeing your revised manuscript.

·

Basic reporting

The manuscript has a clear goal: the description of a new species. The authors do so by combining morphological and genetic data, and compare their specimens with the type specimens of a closely allied species.

Per the "best practices" of peerJ, consider mentioning the name of new species in the title

The abstract needs to be improved grammatically, e.g. the first two sentences of the abstract are difficult to comprehend. Some of the sentences could be split into several shorter sentences as well. The discussion needs to be checked carefully for grammatical mistakes. Some examples are in lines 380, 396, 405.

The references are sufficient, the article structure is professional.

Table 1 is too large and hard to read, at least in its current form. Maybe move to the supplement, or try to turn it into a graph, e.g. heat map?

Figures are mostly well-constructed. I have a few minor issues, which I describe in the following:

Figure 1: species names and labels in general are pretty small and hard to read. Please enlarge them. Also, the colors are difficult against the map background. Maybe add a white background and use black and red font?

Figure 2: the abbreviations “RT” and “AP" in E and F need to be explained in the caption

Figure 5: It is not clear why some specimens are summarised (+ 2 sequences etc). Could this be handled differently? It looks confusing. I suggest you either write down all specimen names, or omit most of them? If you omit most of them, you could label the “GenBank” tips with the species names you use on the side, and write down all of those specimens in the supplementary table. And only use individual labels for the specimens you sequenced yourself. This is not a “you have to do it this way” suggestion, but I do suggest you make this interesting figure clearer. The side labels are also not very clear. Could this become a table maybe? Or could you omit the countries of collection, as you have this info in figure 9?

Figure 8: under A, you write “Creusia spinuloda var. 1 sensu Darwin…”. It should be “Creusia spinulosa var. 1 sensu Darwin…”

Experimental design

Experimental design is mostly adequate. I only have one issue concerning the origin of the sequence data:

Lines 158-170: Do I understand correctly that the sequences were generated in Chan et al. 2018? If so, you should remove this section and refer to Chan et al. 2018 for the origin of the sequence data.

Validity of the findings

Overall, the link between sequences and morphology needs to be clarified. I think the authors had access to several of the specimens from which they used the GenBank sequence data. This is important to know because it increases my confidence in a correct morphological species identification. In general, this is a big issue when using GenBank data.

In addition, I have a few specific issues:
Lines 206-211: I think the parentheses are misplaced. As is, the sentences are partial. But they are all part of the bottles’ label? Instead of repeating the full label, consider introducing an abbreviation in the M&M section for each of the five bottles, such as “(hereafter referred to as the “Amboina” sample)” or similar.
Line 229: remove “and used”. Add a sentence detailing the percentage of missing data per marker.
Lines 239-244: I think this refers to the sequences of your new species.
Line 256: should it be: “with high sequence divergence”? Otherwise please rephrase, this sentence and the following seem to contradict each other.
Lines 259-261: How confident are you that the GenBank sequences were correctly identified?

Additional comments

My general comments concern mostly grammatical suggestions for the introduction.

In several instances, you write “the GenBank”. I would omit “the”.

Lines 57-58: what do you mean by lifestyle modes?
Line 99: replace “barnacle species did not match all available GenBank sequences” with “barnacle species did not match any available GenBank sequence”
Line 115: replace “euspinulosum’ from GenBank sequences “ with “euspinulosum’ entries from GenBank sequences “
Lines 117-118: Replace “As part of study, a mini-review on global distribution of Cantellius species was presented and discussed in aspects of management and conservation of the species diversity. “ with “As part of the study, a mini-review on global distribution of Cantellius species is presented and discussed with regard to management and conservation of the species diversity. “

Reviewer 2 ·

Basic reporting

.

Experimental design

.

Validity of the findings

.

Additional comments

Review
Hyun Kyong Kim et al. DNA-based diversity assessment reveals a new coral barnacle
(Balanomorpha: Pyrgomatidae) exclusively associated with the high latitude coral Alveopora japonica in the waters of southern Korea


One of the achievements of the present study is the retrieval of more than century old original material (Mortensen’s Pacific Expedition 1914–16), used by Broch for the description of C. euspinulosum, and comparing these to the new material using SEM techniques (figs. 3-4). These data are compared to the original graphic material (fig 8). The added to their data the description of the arthropodal characters which are missing in many if not most of the coral barnacles.
The general concept is that Cantellius is a generalist coral barnacle and there is no host specificity. It is the late W.A. Newman who proposed that host specificity is common within the coral inhabiting barnacles. (Hoekia in Hydnophora, Hiroa in Astreopra and more). This concept was followed by other students of Cirripedia.
The authors used molecular techniques to support their hypothesis that Cantellius from Alveopora is a monophyletic clade that currently is known from a single host species. The molecular analyses justify that conclusion.

My conclusion is that the paper should be accepted with mini-minor revision.

In the legend to fig 5 the author should indicate that the phylogenetic tree is based on concatenated segment of 12S and COI.
I suggest that the authors will present the pairwise distances of the two molecular markers between the new species and the sequences from other barnacles morphologicaly identified as C. euspinulosum. In addition they will indicate, in the text, what is the range of pairwise distances within C. alveoporae specimens
Using the Zoobank data presented in the MS I could not find the new species. I am not familiar with this tool and it is possible that the data are released only after the paper is published.

---

## Round 0.2 · accepted · Accept

Thank you for your revised manuscript which I agree has addressed all the concerns raised by the referees in their initial review of your submission. The referees were already very positive about your initial submission, and I find that it is improved by your revisions. Therefore, I am happy to accept your revised manuscript and move it forward into production.